# Parkin Inhibits RANKL-Induced Osteoclastogenesis and Ovariectomy-Induced Bone Loss

**DOI:** 10.3390/biom12111602

**Published:** 2022-10-31

**Authors:** Weiduo Hou, Mo Chen, Chenyi Ye, Erman Chen, Weixu Li, Wei Zhang

**Affiliations:** 1Department of Orthopedics, Second Affiliated Hospital, School of Medicine, Zhejiang University, Hangzhou 310009, China; 2Research Institute of Orthopedics, Zhejiang University, Hangzhou 310009, China; 3Department of Rheumatology, Second Affiliated Hospital, School of Medicine, Zhejiang University, Hangzhou 310009, China

**Keywords:** Parkin, bone loss, osteoclastogenesis, ovariectomy-induced osteoporosis in murine model

## Abstract

Osteoporosis and osteoporotic fractures comprise a substantial health and socioeconomic burden. The leading cause of osteoporosis is an imbalance in bone formation and bone resorption caused by hyperactive osteoclasts. Therefore, a new strategy to suppress osteoclastogenesis is needed. Parkin is likely closely associated with bone metabolism, although its role in osteoclastogenesis is unclear. In this study, the Parkin protein inhibited the receptor activator of nuclear factor-κB ligand (RANKL)-induced osteoclast formation, osteoclast-specific gene expression, F-actin ring formation, and bone resorption pit formation in vitro. Moreover, depletion of Parkin enhanced RANKL-induced osteoclast formation, osteoclast-specific gene expression, F-actin ring formation, and bone resorption pit formation. Reactive oxygen species (ROS) activity was suppressed, while autophagy was upregulated with the presence of the Parkin protein. ROS activity was upregulated and autophagy was decreased due to Parkin knockdown. In addition, intravenous administration of Parkin rescued ovariectomy-induced bone loss and reduced osteoclastogenesis in vivo. Collectively, Parkin has therapeutic potential for diseases associated with overactive osteoclasts.

## 1. Introduction

There are 8.9 million osteoporotic fractures worldwide annually, resulting in substantial social and economic burdens [1]. Approximately half of women will experience an osteoporotic fracture in their lifetime [2]. Therefore, new strategies are needed to prevent osteoporosis. Bone homeostasis is maintained by balancing bone formation and bone resorption; the primary function of osteoclasts is bone resorption. Postmenopausal osteoporosis is characterized by increased osteoclast number and activity [3].

Parkin is encoded by Parkinson juvenile disease protein 2 (PARK2) [4]. Mutations in the PARK2 gene can result in autosomal recessive juvenile Parkinson disease [5,6]. Parkin is expressed in the heart, testis, brain, skeletal muscle, and serum [7]. Notably, Parkin eliminates dysfunctional mitochondria in mammalian cells [8]; mitochondrial dysregulation contributes to bone metabolism disorders [9]. Parkin enhances autophagy and implicates a failure to eliminate dysfunctional mitochondria in the pathogenesis of Parkinson’s disease [10]. Furthermore, loss of Parkin results in excessive inflammatory cytokines [11]. Depletion of Parkin impairs autophagy and increases oxidative stress due to the accumulation of dysfunctional mitochondria that release ROS [12,13]. In adipose-derived mesenchymal stem cells, osteogenic differentiation was reportedly reduced by downregulation of Parkin [8]. Moreover, Parkin has been shown to accelerate stem cell osteogenesis [14]. Osteoclasts are crucial in the development and progress of osteoporosis. However, it is unclear whether Parkin participates in osteoclastogenesis.

Monocytes differentiate into mature osteoclasts upon stimulation by macrophage colony-stimulating factor (M-CSF) and receptor activator of nuclear factor-κB (NF-κB) ligand (RANKL) [15]. RANKL is essential for osteoclast differentiation, activation, and survival [16]. The receptor activator of nuclear factor-κB (RANK), a cell-surface receptor, interacts with RANKL to mediate the induction of osteoclastic markers such as tartrate-resistant acid phosphatase (TRAP), c-FOS, nuclear factor of activated T cells cytoplasmic 1 (NFATc1), and cathepsin K [17]. Moreover, excessive amounts of ROS accelerate osteoclastic differentiation [18,19]. Lee et al. suggested that ROS, the production of which is mediated by RANKL, acts as an intracellular signal for osteoclastogenesis [20]. Estrogen deficiency results in excessive ROS, thereby enhancing osteoclastic bone resorption in postmenopausal osteoporosis [21]. Lin et al. reported that Parkin prevents apoptosis and tissue damage by reducing the concentration of ROS [22]. Furthermore, Ansari et al. revealed that Parkin decreases ROS levels and inhibits chondrocyte apoptosis [23]. In addition, with crucial roles in the maintenance of cellular homeostasis and organ function, autophagy has emerged as a potential target for the prevention and treatment of osteoporosis [24]. Autophagy can mediate osteoclast formation and function [25]. Thus, in this study, we investigated the role of Parkin in osteoclastogenesis using both in vitro and in vivo methods. The acute effects of menopause are modeled by ovariectomy stimulates bone resorption by increasing osteoclast formation and activity [26,27]. Thus, ovariectomy-induced osteoporosis in mice was used in this study. The results reveal that Parkin inhibited RANKL-induced osteoclastogenesis and ovariectomy-induced bone loss.

## 2. Materials and Methods

### 2.1. Reagents

Recombinant Parkin was purchased from R&D Systems. α-minimum essential medium and fetal bovine serum were supplied by Gibco. Recombinant M-CSF and RANKL were purchased from Novoprotein. Primary antibodies against glyceraldehyde-3-phosphate dehydrogenase (GAPDH), autophagy-related 7 (ATG7), and microtubule-associated protein 1 light chain 3 (LC3) were obtained from Proteintech (Wuhan, Hubei, China). Cell Counting Kit-8 (CCK-8) reagents were purchased from Dojindo (Kumamoto, Japan). According to our previous report [14], a siRNA sequence targeted to Parkin (sense strand: 5′-CCUUCUGCCGGGAAUGUAATT-3′, antisense strand: 5′-UUACAUUCCCGGCAGAAGGTT-3′) and a negative control siRNA (sense strand: 5′-UUCUCCGAACGUGUCACGUTT-3′ and antisense strand: 5′-ACGUGACACGUUCGGAGAATT-3′) were purchased from GenePharma Inc. (Shanghai, China).

### 2.2. Bone-Marrow-Derived Monocyte Isolation and Osteoclast Differentiation

This study was approved by the Ethics Committee of the Second Affiliated Hospital, School of Medicine, Zhejiang University; it was performed in accordance with laboratory animal care and use guidelines. Bone-marrow-derived monocytes (BMMs) were isolated from C57BL/6 mice and cultured in α-minimum essential medium with 10% fetal bovine serum [28]. Osteoclast differentiation was induced by M-CSF (30 ng/mL) and RANKL (100 ng/mL; Novoprotein). TRAP staining was performed to assess osteoclast formation. TRAP-positive multinucleated cells (≥3 nuclei) were scored as osteoclasts [18].

### 2.3. Cell Counting Kit-8 Assay

To assess the effect of Parkin on cytotoxicity, BMMs were seeded into 96-well plates (8 × 10^3^ cells/well) and treated with Parkin for 48 or 96 h. Next, BMMs were incubated in 10% CCK-8 solution (Dojindo, Kumamoto, Japan) for 3 h at 37 °C; the absorbance at 450 nm was determined using a microplate reader.

### 2.4. F-Actin Ring Formation Analysis

The effect of Parkin on F-actin ring formation was analyzed by staining with rhodamine-conjugated phalloidin (Invitrogen, Carlsbad, CA, USA). BMMs were seeded into 96-well plates (8 × 10^3^ cells/well) and incubated with serial concentrations of Parkin or siRNA and fresh osteoclastogenesis induction medium containing fresh osteoclastogenic medium for 5 days. Next, 4% paraformaldehyde (Sigma-Aldrich, Shanghai, China) and 0.1% Triton X-100 were applied to fix and permeabilize the cells, respectively. Cells were stained with rhodamine-conjugated phalloidin in 2% bovine serum albumin (1: 200) for 1 h. Nuclei were visualized by 4′,6-diamidino-2-phenylindole (DAPI) staining for 5 min. Fluorescence images of F-actin were acquired using a fluorescence microscope (Leica, Munich, Germany). Three fields were randomly captured, and ImageJ software (NIH) was used to quantify the sizes of F-actin rings.

### 2.5. Osteoclastic Bone Resorption Pit Assay

The effects of Parkin on in vitro osteoclastic bone resorption were assessed using bovine bone slices. Equal numbers of BMMs were seeded onto bone slices in 24-well plates and incubated with Parkin (0, 100, 300, 500 ng/mL) or siRNA, together with RANKL (100 ng/mL) and M-CSF (30 ng/mL), for 10–12 days [28]. Adhered cells were removed by brushing. Bone resorption pits were imaged using a scanning electron microscope (Hitachi S-3700N, Chiyoda, Japan).

### 2.6. Quantitative Real-Time Polymerase Chain Reaction

Following treatment, cells were lysed. Total RNA was extracted as described previously [26]. Real-time polymerase chain reaction analysis was performed using the SYBR^®^ Green PCR Kit (TaKaRa) on the ABI StepOnePlus System™ (Applied Biosystems, Warrington, UK). The cycling conditions were implemented as described previously [28]. Primers were designed and synthetized by Sangon Biotech (Shanghai, China). GAPDH was used as a housekeeping gene. The 2^−^**^ΔΔ^**^Ct^ method was applied to calculate gene expression. Primer sequences are listed in Table 1.

### 2.7. Western Blot Analysis

To determine the effect of Parkin on osteoclast-related protein markers, BMMs (1.2 × 10^5^ cells/well, six-well plates) were cultured with 0 or 300 ng/mL Parkin for 0, 3, and 5 days. To uncover the underlying molecular signaling pathways, BMMs (8 × 10^5^ cells/well, six-well plates) were pretreated with Parkin (300 ng/mL) for 6 h or siRNA (3 days); they were then exposed to RANKL (100 ng/mL) for 0, 5, 10, 15, 30, or 60 min. Cells were lysed in radioimmunoprecipitation assay buffer (Beyotime Biotechnology, Shanghai, China), and the protein lysate was separated by 8–12% sodium dodecyl sulfate–polyacrylamide gel electrophoresis. Subsequently, proteins were electroblotted onto polyvinylidene difluoride membranes (Millipore, Shanghai, China), which were blocked with 5% bovine serum albumin for 1 h, then incubated with primary antibodies at 4 °C for 12 h. The membranes were incubated with the corresponding secondary antibodies (Boster, Wuhan, China) for 2 h at 4 °C. Immunoreactive bands were visualized using enhanced chemiluminescence detection reagent (Millipore) and a chemiluminescence detection system (Bio-Rad XRS, Hercules, CA, USA). 

### 2.8. ROS Detection In Vitro

ROS were measured using an ROS Assay Kit (S0033S, Beyotime Biotechnology) [29,30]. Cells were incubated with 20 μM DCFH-DA for 30 min at 37 °C and fluorescence intensity was assayed by fluorescence microscope.

### 2.9. In Vivo Evaluation

Animal experiments were carried out in accordance with the National Institutes of Health guide for the care and use of Laboratory animals (NIH Publications No. 8023, revised 1978) and the laboratory animal care and use guidelines of the Animal Care and Use Committee of the Second Affiliated Hospital of Zhejiang University School of Medicine. Fifteen female C57BL/6 mice (12 weeks old, approximately 20 g) were provided by Zhejiang Academy of Medical Sciences. They were randomly divided into the following groups (*n* = 5 per group): sham operation (sham), ovariectomy-induced osteoporosis (OVX), and OVX + Parkin. The OVX-induced osteoporosis model was established in accordance with previous studies [28,31]. At 1 week after the operation, mice in the OVX + Parkin group were intraventricularly administered 10 μL of Parkin (dose adjusted to mouse body weight (300 ng/g); once per week for 5 weeks), whereas mice in the sham and OVX groups received an identical volume of phosphate-buffered saline. The mice were euthanized; femur specimens were collected and fixed in 4% paraformaldehyde. Fixed femur specimens were subjected to microcomputed tomography (Scanco Medical, Zurich, Switzerland). The bone volume/tissue volume ratio (BV/TV), trabecular number (Tb.N), and trabecular separation (Tb.Sp) were calculated. 

Fixed femur specimens were decalcified in 10% ethylenediaminetetraacetic acid (Sigma-Aldrich, Shanghai, China) for 4 weeks, embedded, and cut into serial sections of 5 μm thickness. Histomorphologic evaluation was carried out by hematoxylin and eosin (H&E), Masson’s trichrome (Nanjing Jiancheng Bioengineering Institute; Nanjing, China), and TRAP staining.

### 2.10. Statistical Analysis

Experiments were performed at least three times. Continuous data are shown as means ± standard deviations. Statistical analyses were carried out using SPSS software (IBM, Armonk, IL, USA). Differences between groups were compared by two-tailed Student’s *t*-test or one-way analysis of variance. *p* values < 0.05 were considered indicative of statistical significance.

## 3. Results

### 3.1. Parkin Expression Gradually Decreased during Osteoclastic Differentiation and Downregulation of PARKIN by siRNA Increased Osteoclastogenesis in BMMs 

To investigate the levels of Parkin in osteoclast differentiation, WB analysis was performed at 0, 1, and 3 days after the induction. Protein expression of Parkin gradually decreased during osteoclastogenesis (Figure 1A).

To assess the role of endogenous Parkin in osteoclastic differentiation, the expression of Parkin was downregulated by siRNA. After siRNA transfection, the expression of Parkin was examined by PCR. Expression of Parkin was downregulated by siRNA, especially siRNA3 (Appendix A). Thus, siRNA3 was selected for subsequent experiments. The TRAP staining showed more osteoclast formation in the Parkin siRNA group. Quantitative results reveal that Parkin-siRNA treatment increased the number of TRAP-positive cells by about 2.97-fold (Figure 1B). Moreover, F-actin rings were much larger when Parkin was silenced using siRNA (Figure 1C). Parkin knockdown also increased the number of bone resorption pits. Quantitative analysis showed a 4.5-fold increase in the Parkin-siRNA group (Figure 1D). Quantitative analysis of Western blot revealed that, compared with the control group (scramble), Parkin-siRNA reduced the protein levels of Parkin by 54 and 63% on days 1 and 3 during the osteoclast differentiation, respectively. Compared with the scramble group, the protein levels of c-Fos and NFATc1 were increased by 1.5- and 2.3-fold on day 1 in the Parkin-siRNA group, respectively. Significant upregulation of c-Fos and NFATc1 were also found on day 3 after Parkin-siRNA treatment (Figure 1E).

### 3.2. Parkin Protein Suppresses Osteoclast Formation and Bone Resorption In Vitro

Firstly, the impacts of Parkin on cell viability were assessed by CCK-8 assay. Cell viability was not significantly affected by 48 and 96 h of Parkin treatment (0, 0.1, 1, 100, 500, and 1000 ng/mL) (Figure 2A,B). Then, to investigate the effect of Parkin on the osteoclast formation, a TRAP staining was performed. After exposing the different concentration of Parkin (0, 100, 300, and 500 ng/mL) during the induction of osteoclastic differentiation for 5 days, the osteoclasts were counted. The results reveal that Parkin protein suppressed osteoclast formation in a dose-dependent manner (*p* < 0.05, Figure 2C,D). To investigate the role of Parkin on the function of osteoclast bone resorption further, a bone resorption pit assay was also carried out. Bone resorption pit assays showed that Parkin (0, 100, 300, and 500 ng/mL) significantly decreased the size and number of pits (Figure 2E,F).

### 3.3. Parkin Protein Inhibits Osteoclast-Specific Genes and F-Actin Ring Formation

Specific genes during osteoclastic differentiation (ATPase, MMP9, NFATc1, and TRAP) were downregulated by Parkin on days 3 and 5 (Figure 3A–D). Moreover, c-Fos and NFATc1 protein levels in RANKL-induced osteoclasts were decreased by Parkin (300 ng/mL) on days 0, 3, and 5 (Figure 3E–G). Osteoclast function requires extensive F-actin reorganization, which is a landmark of osteoclast [32]. As shown in Figure 3H, F-actin rings were smaller after Parkin protein treatment (300 ng/mL). Quantification analysis showed that compared with cells without Parkin, the average size of F-actin rings declined by 63% in the Parkin treatment group. (*p* < 0.05, Figure 3I).

### 3.4. Parkin Protein Increases Autophagy and Decreases RANKL-Induced ROS Levels, While Parkin Knockdown Decreases Autophagy and Increases RANKL-Induced ROS Levels during Osteoclastogenesis in BMMs

We next assayed the levels of autophagy and ROS. Firstly, the optimal concentration of Parkin to increase autophagy marks was investigated. A 300 ng/mL concentration was found to be optimal (Appendix A). In the autophagy signaling pathway, the ATG7 protein and LC3 II/I ratio were increased by Parkin treatment at 10, 15, 30, and 60 min (Figure 4A–C). We also evaluated the effects of Parkin on RANKL-induced ROS levels during osteoclastogenesis. There was a robust decrease at 100 and 300 ng/mL Parkin (60% and 80%, respectively) in ROS levels (Figure 4D,E). Moreover, the levels of autophagy and ROS were also investigated when Parkin was downregulated by siRNA. Parkin depletion attenuated the levels of autophagy (ATG7 and LC3 II/I) (Figure 4F–H). The fluorescence of ROS activity was significantly upregulated due to Parkin knockdown (Figure 4I,J).

### 3.5. Parkin Delayed the Progression of OVX-Induced Osteoporosis In Vivo

To identify the effect of Parkin on bone loss in vivo, OVX-induced osteoporosis was established. Microcomputed tomography showed that OVX induced significant trabecular bone mass decrease, which was significantly rescued by Parkin (Figure 5A). According to the quantitative analysis, the OVX+Parkin group displayed a significantly higher extent of trabecular bone mass (BV/TV  =  0.35  ±  0.04) than the OVX group (BV/TV = 0.23  ±  0.06). The sham group showed the highest bone volume. These results indicate that OVX resulted in excessive bone loss, which was increased to a certain extent by intravenous injection of Parkin. Histological analysis showed a marked bone mass decrease in the OVX group, which was significantly rescued by Parkin treatment (Figure 5F). The quantitative results of Masson staining showed that the OVX+Parkin group exhibited about 2-fold increase in BV/TV compared with the OVX group (Figure 5G). In addition, the TRAP staining revealed that Parkin decreased the number of osteoclasts (Figure 5F,H).

## 4. Discussion

Osteoclasts secrete acid and lytic enzymes to degrade bone matrix [33], a process essential for the maintenance of bone metabolism balance [15]. Greater bone resorption than bone formation results in osteoporosis. Therefore, suppression of osteoclastic differentiation and activity could delay osteoporosis progression. Several drug treatments are available, including bisphosphonates, estrogen or estrogen-replacement therapy, and denosumab. However, these agents have side effects, including atypical femur facture and osteonecrosis [34]. Therefore, innovative therapeutic strategies are needed for osteoporosis. To our knowledge, this is the first study of the role of Parkin in osteoclastogenesis. The Parkin protein inhibited osteoclastic differentiation, and downregulating Parkin enhances osteoclastic differentiation. The Parkin protein repressed ROS activity and upregulated autophagy. Moreover, Parkin knockdown increased ROS and suppressed autophagy signaling. Additionally, the Parkin protein delayed the progression of OVX-induced osteoporosis in mice. Herein, Parkin might be considered as an innovative therapeutic target to inhibit osteoclasts by suppressing ROS and autophagy signaling. (Figure 6).

Osteoclasts are multinucleated macrophages derived from monocyte/macrophage precursors. RANKL binds with RANK on the surfaces of osteoclasts and precursors to activate downstream signaling pathways to promote osteoclast differentiation and function [35]. Osteoclasts express markers including TRAP, MMP-9, cathepsin K, NFATc1, ATPase, and c-Fos [36]. Following RANKL induction, c-Fos binds to the NFATc1 promoter to induce autoamplification during osteoclast differentiation [37]. NFATc1 is a master transcription factor that regulates the expression of osteoclast-specific genes such as MMP-9, TRAP, cathepsin K, and ATPase [38]. In this study, Parkin significantly decreased the expression levels of TRAP, MMP-9, cathepsin K, NFATc1, ATPase, and c-Fos. Furthermore, Parkin significantly reduced the number of TRAP-positive cells, decreased the size of F-actin rings, and reduced the number of bone resorption pits. 

Autophagy dysregulation results in osteoclast activation and accelerated bone loss [25]. LC3 conversion (LC3 I to LC3 II) is a characteristic of autophagy [39]. ATG7 is essential for autophagy activation and accelerates the conversion of LC3 I to LC3 II during autophagy [39,40]. Previous studies demonstrated that autophagy-related proteins were significantly downregulated during the course of osteoclastic differentiation [41]. Autophagy was activated in the osteoclasts of human rheumatoid arthritis by upregulating ATG7, and ATG7 overexpression decreased osteoclastogenesis and TNFα-induced bone erosion [42]. Parkin triggers autophagy by ubiquitinating outer mitochondrial membrane proteins [43]. Many studies revealed that Parkin-mediated autophagy can inhibit ROS activity to maintain cellular homeostasis [7,22]. Lin et al. suggested that Parkin-induced autophagy protected against contrast-induced acute kidney injury via decreasing ROS and NLRP3 inflammasome activation [22]. Parkin overexpression was sufficient to induce mitophagy to attenuate ROS and cellular senescence [44]. As essential intracellular secondary messengers, ROS have diverse functions in proliferation, differentiation, and apoptosis [30]; they also drive osteoclastogenesis [45]. ROS accelerate osteoclast formation and function, and ROS-mediated impairment of the antioxidant pathway increases bone loss [19]. Here, we found that Parkin increased the LC3II/I ratio and ATG7 levels. Therefore, Parkin significantly enhanced autophagy during osteoclastogenesis. Parkin also downregulated ROS activity in this study. Therefore, we speculate that ROS activity was inhibited by Parkin-mediated autophagy during osteoclastogenesis. 

This study had several limitations. First, it preliminarily validated the effectiveness of recombinant Parkin, but the safety of this agent should be verified. Second, the findings should be confirmed in transgenic or knockout mice. 

## 5. Conclusions

Parkin protein suppressed osteoclastogenesis in vitro and arrested OVX-induced osteoporosis progression. The downregulation of Parkin enhanced RANKL-induced osteoclastogenesis. We inferred that Parkin may have therapeutic potential for osteoclast-related disorders. Moreover, Parkin disfunction increases inflammation and development of Parkinson’s disease [46]. Interestingly, patients with Parkinson’s disease have a higher risk of osteoporotic fracture [47], which indicated that Parkin may be an intrinsic link between these two diseases. In addition, Parkin may be considered as a useful clinical biomarker for osteoporosis.

## Figures and Tables

**Figure 1 biomolecules-12-01602-f001:**
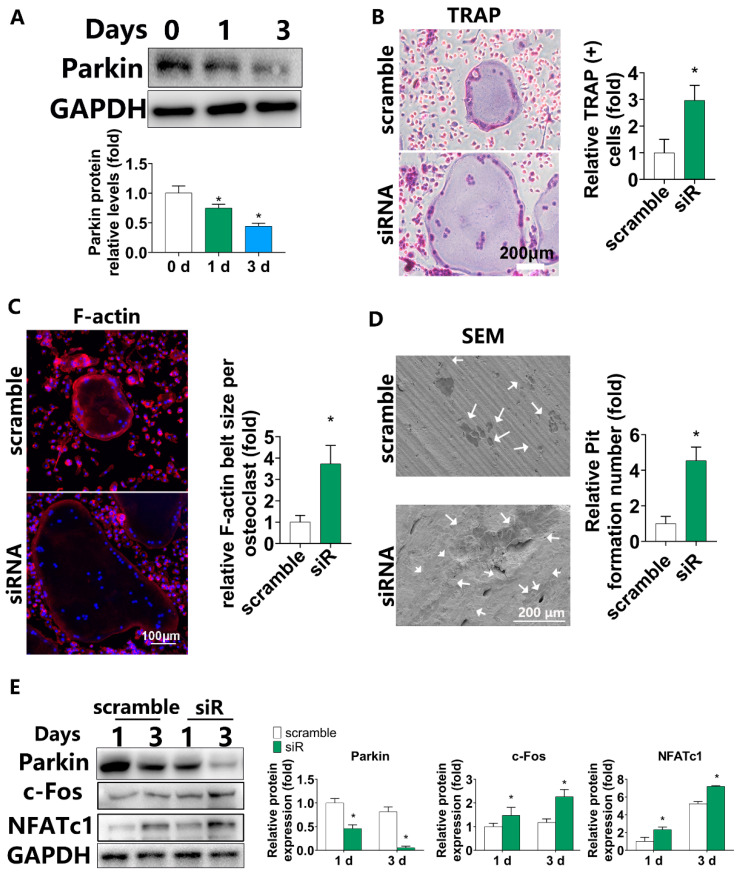
Parkin expression gradually decreases during osteoclastic differentiation and Parkin depletion enhances osteoclast differentiation in BMM. (**A**) Western blots showing the expression of Parkin at days 0, 1, and 3 days after osteoclastic induction. (**B**) One day after siRNA transfection, the cells were treated with RANKL (100 ng/mL) and M-CSF (30 ng/mL) for 5 days. The cells were fixed and stained for TRAP. “Scramble” represents BMMs transfected with control siRNA. “siRNA” represents BMMs transfected with Parkin-siRNA. Scale bars = 200 µm. * *p* < 0.05 compared with scramble group. (**C**) One day after siRNA transfection, the cells were treated with RANKL (100 ng/mL) and M-CSF (30 ng/mL) for 5 days. The cells were fixed and stained with F-actin ring formation analysis. “Scramble” represents BMMs transfected with control siRNA. “siRNA” represents BMMs transfected with Parkin-siRNA. Scale bars = 100 µm. * *p* < 0.05 compared with scramble group. (**D**) BMMs transfected with control or Parkin-siRNAs were induced with M-CSF and RANKL for 10 days. Representative SEM (scanning electron microscopy) images for bone resorption pits. “Scramble” represents BMMs transfected with control siRNA. “siRNA” represents BMMs transfected with Parkin-siRNA. White arrow represents pit formation. (**E**) Parkin, NFATc1, and c-Fos protein levels were analyzed by Western blot analysis (1 and 3 days of osteoclastogenesis). “Scramble” represents BMMs transfected with control siRNA. “siR” represents BMMs transfected with Parkin-siRNA. * *p* < 0.05 compared with scramble group.

**Figure 2 biomolecules-12-01602-f002:**
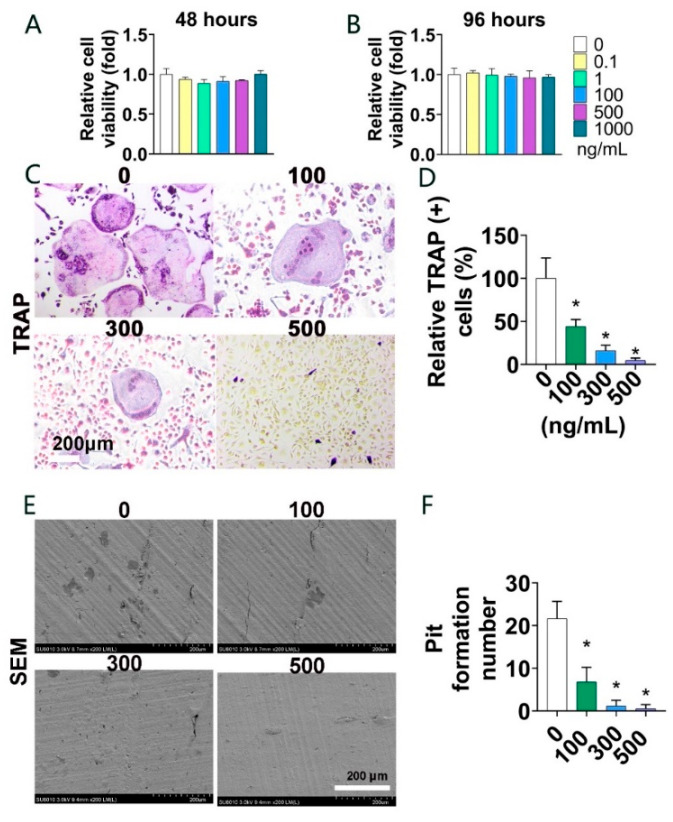
Parkin suppressed osteoclast formation in vitro. (**A**,**B**) CCK-8 assays showed no cytotoxic effect of Parkin (0–1000 ng/mL) on BMMs after treatment for 48 and 96 h. (**C**,**D**) BMMs were treated with RANKL (100 ng/mL) and M-CSF (30 ng/mL) in the presence of 0, 100, 300, and 500 ng/mL Parkin for 5 days. TRAP staining showed that Parkin reduced osteoclast number and size in a dose-dependent manner. Scale bars = 200 µm. * *p* < 0.05 compared with the absence of Parkin (0 ng/mL). (**E**,**F**) Representative SEM (scanning electron microscopy) images showed that Parkin significantly reduced the size of resorption pits in a dose-dependent manner. BMMs were treated with RANKL (100 ng/mL) and M-CSF (30 ng/mL) in the presence of 0, 100, 300, and 500 ng/mL Parkin during osteoclast induction for 5 days. * *p* < 0.05 compared with the absence of Parkin (0 ng/mL).

**Figure 3 biomolecules-12-01602-f003:**
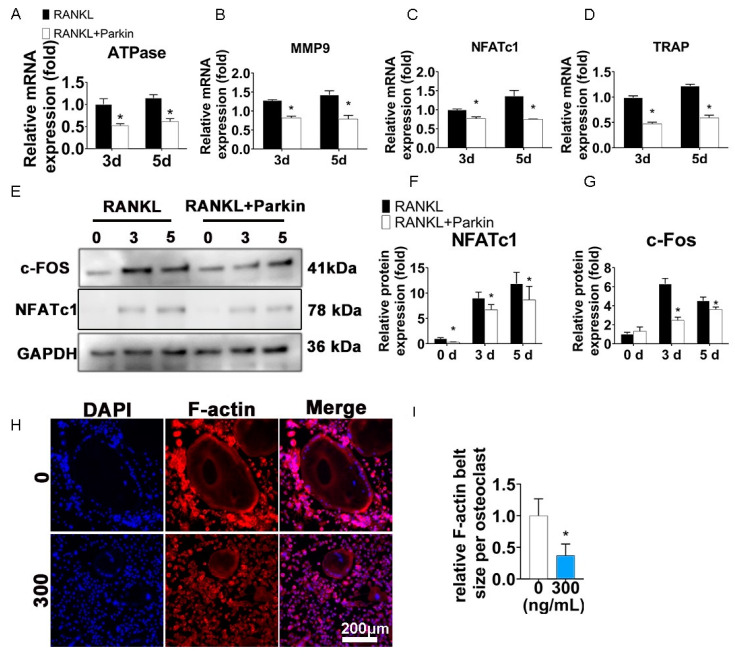
Parkin attenuates osteoclast-specific markers and RANKL-induced F-actin ring formation in vitro. (**A**–**D**) After BMMs were exposed to RANKL (100 ng/mL), M-CSF (30 ng/mL), and Parkin (300 ng/mL) for 3 or 5 days, the mRNA levels of osteoclast-specific genes were significantly suppressed by Parkin. (**E**–**G**) Parkin suppressed NFATc1 and c-Fos protein levels (0, 3, and 5 days of osteoclastogenesis). After BMMs were exposed to Parkin (300 ng/mL) for 1 day without RANKL and MCS-F, the NFATc1 results of WB at day 0 were analyzed. * *p* < 0.05 compared with the control group (0 ng/mL Parkin). (**H**,**I**) F-actin ring staining for osteoclasts. BMMs were treated with Parkin (300 ng/mL) during osteoclastogenesis for 5 days. * *p* < 0.05 compared with the control group (0 ng/mL Parkin).

**Figure 4 biomolecules-12-01602-f004:**
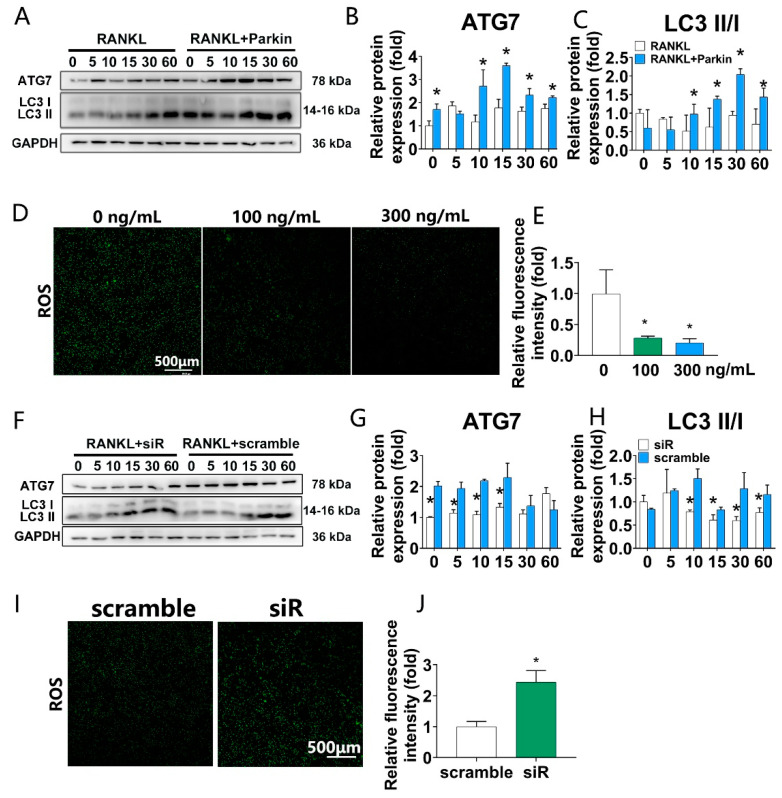
Parkin regulates autophagy and ROS levels during osteoclastogenesis. (**A**–**C**) BMMs were pre-treated with vehicle or Parkin (300 ng/mL) for 6 h and thereafter exposed to RANKL (100 ng/mL) for indicated time (0, 5, 10, 15, 30, and 60 min). Western blots showing that Parkin (300 ng/mL) significantly increased autophagy signaling (ATG7 and LC3II/I). * *p* < 0.05 compared with the control group (0 ng/mL Parkin). (**D**,**E**) BMMs were pre-treated with vehicle or Parkin (300 ng/mL) for 6 h and thereafter exposed to RANKL (100 ng/mL) for 1 day. DCFH-DA fluorescence analysis of ROS. * *p* < 0.05 compared with the control group (0 ng/mL Parkin). (**F**–**H**) BMMs were pre-treated with control siRNA(scramble) or Parkin siRNA (siRNA) for 1 day and then exposed to RANKL (100 ng/mL) for indicated time (0, 5, 10, 15, 30, and 60 min). Western blots showing that Parkin siRNA significantly decreased autophagy signaling (ATG7 and LC3II/I). (**I**,**J**) BMMs were pre-treated with control siRNA(scramble) or Parkin-siRNA (siRNA) for 1 day and then exposed to RANKL (100 ng/mL) for 1 day. * *p* < 0.05 compared with the control group (scramble).

**Figure 5 biomolecules-12-01602-f005:**
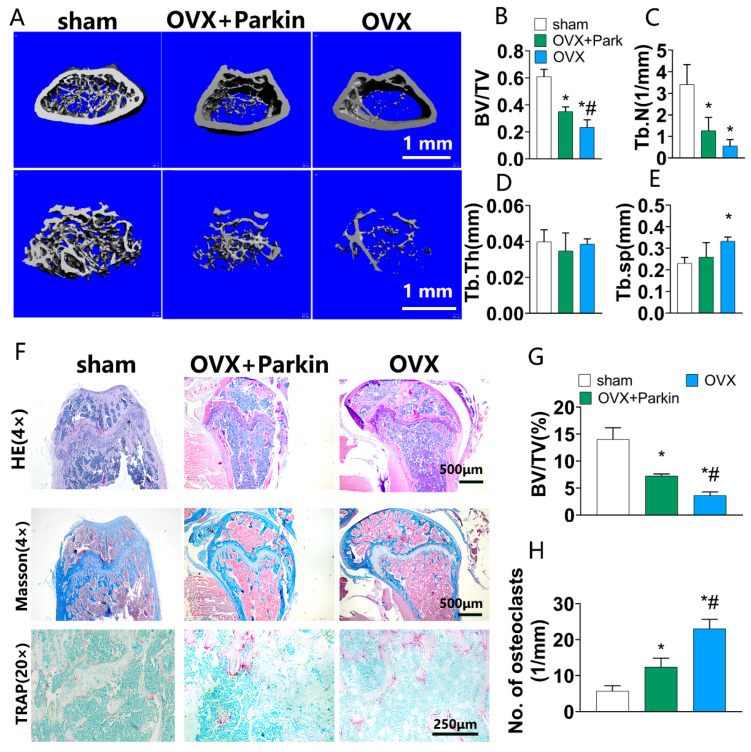
Parkin delayed the progression of OVX-induced bone loss in vivo. (**A**) Representative microcomputed tomography images of mouse distal femur. Scale bars = 1 mm. (**B**–**E**) Quantification of BV/TV, Tb.N, Tb.Th, and Tb.Sp. (**F**) Histologic analyses by H&E, Masson’s, and TRAP staining. (**G**) BV/TV of sections quantified using ImageJ. * *p* < 0.05 compared with the sham group, # *p* < 0.05 compared with the OVX group. (**H**) Number of TRAP (+) cells. * *p* < 0.05 compared with the sham group; # *p* < 0.05 compared with the OVX group.

**Figure 6 biomolecules-12-01602-f006:**
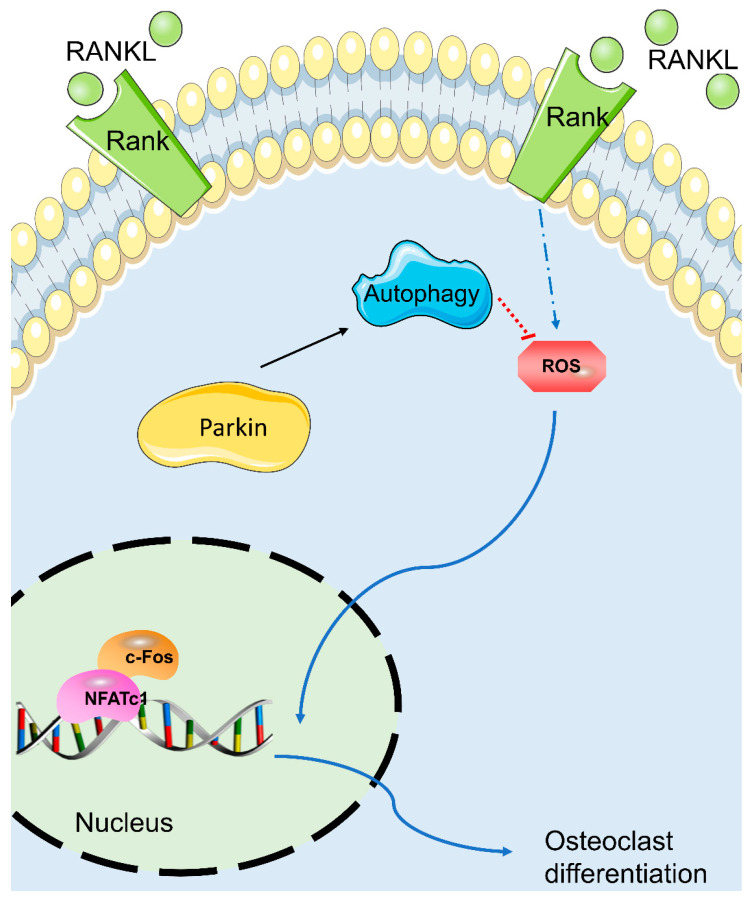
Schematic depicting the putative mechanism by which Parkin induces osteoclast differentiation.

**Table 1 biomolecules-12-01602-t001:** Sequences of primers for PCR analysis.

Gene	Reverse (5′-3′)	Reverse (3′-5′)
TRAP	CACTCCCACCCTGAGATTTGT	CCCCAGAGACATGATGAAGTCA
NFATc1	GGAGAGTCCGAGAATCGAGAT	TTGCAGCTAGGAAGTACGTCT
ATPase	CAGAGCTGTACTTCAATGTGGAC	AGGTCTCACACTGCACTAGGT
MMP-9	CTGGACAGCCAGACACTAAAG	CTCGCGGCAAGTCTTCAGAG
GAPDH	CGACTTCAACAGCAACTCCCACTCTTCC	TGGGTGGTCCAGGGTTTCTTACTCCTT

## Data Availability

All datasets presented in this study are included in the article.

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
