# Peer review of "Parkin Inhibits RANKL-Induced Osteoclastogenesis and Ovariectomy-Induced Bone Loss"

_biomolecules, 2022, doi:10.3390/biom12111602_

Round 1
Reviewer 1 Report
The authors did not mentioned the used animal model in the title neither in the keywords
Acronymes RANKL (line 17) ROS (line 20) PARK2 (line 35) should be explained when mentioned for the first time
IN VIVO and IN VITRO should be in italics
Line 63 - in the goal of the study, the rational of the used model should be described.
Author Response
Reviewer #1:
Thank you very much for giving us the opportunity to revise our manuscript entitled “Parkin inhibits RANKL-induced osteoclastogenesis and ovariectomy-induced bone loss” with the above ID number. We also appreciate your valuable comments and suggestions. Accordingly, we have revised our manuscript and incorporated these comments and suggestions in the revised manuscript. The point-to-point responses to the comments from reviewers are provided as follows. We feel that the manuscript has been significantly improved and hope it is now acceptable by Biomolecules.
In order to facilitate the review process, the new added and changed contents are noted using 'track changes' in WORD for reviewers’ convenience.
The authors did not mentioned the used animal model in the title neither in the keywords.
RE: Thanks for your suggestion. We have added the animal model in the keywords.
Acronymes RANKL (line 17) ROS (line 20) PARK2 (line 35) should be explained when mentioned for the first time. IN VIVO and IN VITRO should be in italics
RE: Thanks for your professional comments. According to your suggestion, we have revised the relevant contents.
Line 63 - in the goal of the study, the rational of the used model should be described.
RE: Thanks for your professional comments. According to your suggestion, we have added the the rational of the used model, as following “The acute effects of menopause are modeled by ovariectomy stimulates bone resorption by increasing osteoclast formation and lifespan (1, 2). Thus, ovariectomy-induced osteoporosis in mice was used in this study.”
References
- Ni S, Yuan Y, Qian Z, Zhong Z, Lv T, Kuang Y, et al. Hypoxia inhibits RANKL-induced ferritinophagy and protects osteoclasts from ferroptosis. Free Radic Biol Med 2021;169,271-82.
- Li JY, Tawfeek H, Bedi B, Yang X, Adams J, Gao KY, et al. Ovariectomy disregulates osteoblast and osteoclast formation through the T-cell receptor CD40 ligand. Proc Natl Acad Sci U S A 2011;108,768-73.
In order to facilitate the review process, the new added and changed contents are noted using 'track changes' in WORD for reviewers’ convenience. We also provided a clean version of the manuscript to cooperate with the editorial office.
The authors did not mentioned the used animal model in the title neither in the keywords.
RE: Thanks for your suggestion. We have added the animal model in the keywords.
Acronymes RANKL (line 17) ROS (line 20) PARK2 (line 35) should be explained when mentioned for the first time. IN VIVO and IN VITRO should be in italics
RE: Thanks for your professional comments. According to your suggestion, we have revised the relevant contents.
Line 63 - in the goal of the study, the rational of the used model should be described.
RE: Thanks for your professional comments. According to your suggestion, we have added the the rational of the used model, as following “The acute effects of menopause are modeled by ovariectomy stimulates bone resorption by increasing osteoclast formation and lifespan [1, 2]. Thus, ovariectomy-induced osteoporosis in mice was used in this study.”

Reviewer 2 Report
Parkin knockout mice lines were generated by different labs with different methods for almost two decades, but very few studies reported these mice to have abnormal bone phenotyping, even in aging Parkin knockout mice. In this manuscript, the authors show that Parkin protein inhibits rankl-induced osteoclastogenesis and ovariectomy-induced bone loss by measuring osteoclast formation, osteoclast-specific gene expression, and bone resorption pit formation in vitro and OVX mice in vivo.
In general, the author still needs some experiments, and high-quality photos to support the conclusion. Especially need high-quality WB photos.
Comments:
1. In 3.1., the Authors knockdown Parkin protein in BMMs by siRNA, but they did not display the efficiency of siRNA, and what is the optimal time to detect Parkin protein level after siRNA transfected.
2. The authors mentioned that "western blot revealed that Parkin siRNA reduced the protein levels of Parkin by 54 and 63% on days 3 and 5 during the osteoclast differentiation, respectively." However, the authors also showed that Parkin protein expression was gradually decreased during osteoclastogenesis, How much percent Parkin protein reduced on day 3 and day 5 during the osteoclastogenesis by itself?
3. The WB figures (figure 1-A and figure 1-E) are of low quality and could not support the results.
4. Figure1-B, C did not match the results mentioned in 3.1. To make sure the figures and the labels are correct.
5. In figure1-D, the bottom photo has too many arrows and it is unclear what the arrow means.
6. In figure 3-F, how long after Parkin treated the BMMs to get 0 day NFATc1 result?
7. In result 3.4, the manuscript needs one figure to show optimal concentration and time for Parkin to increase autophagy marks.
8. In Figures 4-A and 4-F, what is the reason the authors chose the time points (5 min to 60 min after rankl exposure) to measure ATG7 protein and LC3 II/I ratio? Why add rankl to the BMMs? Is rankl reduced BMMs autophagy at that time point? The same, in Figures 4-D and 4-I, why need to add rankl? Is rankl at that time point increase BMMs ROS?
9. In figure 5-B, what is the BV/TV unit?
10. Why Tb. Th (figure 5-D) and Tb.sp (figure 5-E) did not significantly difference between Sham with ovx mice?
Mini comments:
Check the superscript of the number of BMMs amount, for example, in line 93 and line 100, it will be “8x103 cells/well”.
In line 119: total RNA was extracted from which tissue needs to be mentioned in the manuscript.
In 3.5, why use the “delayed” not inhibited or suppressed or another word?
Author Response
Reviewer 2:
Parkin knockout mice lines were generated by different labs with different methods for almost two decades, but very few studies reported these mice to have abnormal bone phenotyping, even in aging Parkin knockout mice. In this manuscript, the authors show that Parkin protein inhibits rankl-induced osteoclastogenesis and ovariectomy-induced bone loss by measuring osteoclast formation, osteoclast-specific gene expression, and bone resorption pit formation in vitro and OVX mice in vivo.
In general, the author still needs some experiments, and high-quality photos to support the conclusion. Especially need high-quality WB photos.
RE: We appreciate you for your valuable comments and suggestions. Accordingly, we have revised our manuscript and incorporated these comments and suggestions in the revised manuscript. The point-to-point responses to the comments are provided as follows. We feel that the manuscript has been significantly improved and hope it is now acceptable by Biomolecules.
In order to facilitate the review process, the new added and changed contents are noted using 'track changes' in WORD for reviewers’ convenience.
Comments:
- In 3.1., the Authors knockdown Parkin protein in BMMs by siRNA, but they did not display the efficiency of siRNA, and what is the optimal time to detect Parkin protein level after siRNA transfected.
RE: Thanks for your professional comments. At day 2 after siRNA transfection, the expression of Parkin was examined by PCR. Expression of Parkin was downregulated by siRNA, especially siRNA3, which was consisted with our previous study[3]. Immediately after, at days 2 and 3 post transfection by siRNA3, the proteins of Parkin were also analyzed. the greatest reduction of Parkin was found at day 2 after transfection, as following:
Supplemental figure 1. A, mRNA levels of Parkin were analyzed by PCR at day 2 after small-interfering RNA treatment. “scramble” represents BMMs transfected with control siRNA. “siR1” represents BMMs transfected with Parkin-siRNA1. “siR2” represents BMMs transfected with Parkin-siRNA2;“siR3” represents BMMs transfected with Parkin-siRNA3; * P < 0.05 compared with scramble group. B, Protein levels of Parkin were analyzed by WB at days 0, 2, 3 after small-interfering RNA3 treatment.
- The authors mentioned that "western blot revealed that Parkin siRNA reduced the protein levels of Parkin by 54 and 63% on days 3 and 5 during the osteoclast differentiation, respectively." However, the authors also showed that Parkin protein expression was gradually decreased during osteoclastogenesis, How much percent Parkin protein reduced on day 3 and day 5 during the osteoclastogenesis by itself?
RE: Thanks for your professional comments. We are very sorry that the meaning of this expression is not clear. Compared with control group (scramble siRNA), Parkin siRNA reduced the protein levels of Parkin by 54 and 63% on days 1 and 3 during the osteoclast differentiation, respectively (Figure 1E). Moreover, compared with undifferentiated cells (0 day), the protein levels of Parkin itself were decreased by 25% and 56% on days 1 and 3 during the osteoclast differentiation, respectively(Figure 1A).
Figure 1 A: Western blots showing the expression of Parkin at days 0, 1, and 3 days after osteoclastic induction.
Figure 1E: Parkin, NFATc1 and c-Fos protein levels were analyzed by Western blot analysis (1 and 3 days of osteoclastogenesis). “scramble” represents BMMs transfected with control siRNA. “siR” represents BMMs transfected with Parkin-siRNA. * P < 0.05 compared with scramble group.
- The WB figures (figure 1-A and figure 1-E) are of low quality and could not support the results.
RE: Thanks for your professional comments. we have re-organized Figure 1, as following:
Figure 1. Parkin expression gradually decreases during osteoclastic differentiation and Parkin depletion enhances osteoclast differentiation in BMM. (A) Western blots showing the expression of Parkin at days 0, 1, and 3 days after osteoclastic induction. (B) One day after siRNA transfection, the cells were treated with RANKL (100 ng/mL) and M-CSF (30 ng/mL) for 5 days. The cells were fixed and stained for TRAP. “scramble” represents BMMs transfected with control siRNA. “siRNA represents” BMMs transfected with Parkin-siRNA. Scale bars = 200 µm. * P < 0.05 compared with scramble group. (C) One day after siRNA transfection, the cells were treated with RANKL (100 ng/mL) and M-CSF (30 ng/mL) for 5 days. The cells were fixed and stained with F-actin ring formation analysis. “scramble” represents BMMs transfected with control siRNA. “siRNA represents” BMMs transfected with Parkin-siRNA. Scale bars = 100 µm. * P < 0.05 compared with scramble group. (D) BMMs transfected with control or Parkin siRNAs were induced with M-CSF and RANKL for 10 days. Representative SEM (scanning electron microscopy) images for bone resorption pits. “scramble” represents BMMs transfected with control siRNA. “siRNA represents” BMMs transfected with Parkin-siRNA. (E) Parkin, NFATc1 and c-Fos protein levels were analyzed by Western blot analysis (1 and 3 days of osteoclastogenesis). “scramble” represents BMMs transfected with control siRNA. “siR” represents BMMs transfected with Parkin-siRNA. * P < 0.05 compared with scramble group.
- Figure1-B, C did not match the results mentioned in 3.1. To make sure the figures and the labels are correct.
RE: Thanks for your professional comments. We are very sorry these errors. We have revised the figure and make sure it.
- In figure1-D, the bottom photo has too many arrows and it is unclear what the arrow means.
RE: Thanks for your professional comments. according to your suggestions, we have delated some arrows. The arrow represents bone resorption pit by osteoclasts.
- In figure 3-F, how long after Parkin treated the BMMs to get 0 day NFATc1 result?
RE: Thanks for your professional comments. After BMMs were exposed to Parkin (300 ng/mL) for 1 day without RANKL and MCS-F, the NFATc1 results of WB at day 0 were analyzed. The relevant figure legends were added in the manuscripts.
- In result 3.4, the manuscript needs one figure to show optimal concentration and time for Parkin to increase autophagy marks.
RE: Thanks for your professional suggestions. according to your suggestion, the optimal concentration of Parkin to increase autophagy marks was investigated, as following figure. A 300 ng/mL concentration was found to be optimal.
Supplemental figure 2. BMMs were pre-treated with vehicle or Parkin (50, 300, 600 ng/mL) for 6 h and thereafter exposed to RANKL (100 ng/mL) for 1 day. Then, Western blots analysis were performed.
- In Figures 4-A and 4-F, what is the reason the authors chose the time points (5 min to 60 min after rankl exposure) to measure ATG7 protein and LC3 II/I ratio? Why add rankl to the BMMs? Is rankl reduced BMMs autophagy at that time point? The same, in Figures 4-D and 4-I, why need to add rankl? Is rankl at that time point increase BMMs ROS?
RE: Thanks for your professional comments. Based on previous studies[4-6], we chose the time points to measure ATG7 protein and LC3 II/I ratio. BMMs were pre-treated with vehicle or Parkin (300 ng/mL) for 6 h and thereafter exposed to RANKL (100 ng/mL) for indicated time (0, 5, 10, 15, 30 and 60 min). The treatment of RANKL is an attempt to mimic a microenvironment of osteoclast differentiation, which is consisted with previous reports[4-7]. Based on previous studies, RANKL will increase BMMs ROS and autophagy[8-10].
- In figure 5-B, what is the BV/TV unit?
RE: Thanks for your professional comments. The BV/TV represents bone volume/tissue volume ratio, thus it has no unit.
- Why Tb. Th (figure 5-D) and Tb.sp (figure 5-E) did not significantly difference between Sham with ovx mice?
RE: Thanks for your professional comments. We are sorry for forgetting the “*”in Tb.sp (figure 5-E), and significant difference were found in Tb.sp between Sham with ovx mice. However, Tb. Th (figure 5-D) Tb.Th showed a decreasing trend in OVX group, but had no difference between Sham and OVX group. Tb. Th (trabecular thickness) indicates new bone formation. Bone resorption is a major manifestation in the OVX model. These results are similar with previous studies [11, 12]. We also revised the errors.
Mini comments:
Check the superscript of the number of BMMs amount, for example, in line 93 and line 100, it will be “8x103 cells/well”.
RE: Thanks for your comments. We have revised these errors.
In line 119: total RNA was extracted from which tissue needs to be mentioned in the manuscript.
RE: Thanks for your professional comments. according to your suggestions, we have revised the manuscript. As following “Following treatment, cells were lysed.”(Line 122)
In 3.5, why use the “delayed” not inhibited or suppressed or another word?
RE: Thanks for your professional comments. The microenvironment for estrogen deficiency-induced osteoporosis is complex, which involves multiple cytokines, cells, and receptors[13]. Our results showed OVX resulted in excessive bone loss, which was increased to a certain extent by the treatment of Parkin. thus, we use “delayed” not inhibited or suppressed.
References
[1] S. Ni, Y. Yuan, Z. Qian, Z. Zhong, T. Lv, Y. Kuang, B. Yu, Hypoxia inhibits RANKL-induced ferritinophagy and protects osteoclasts from ferroptosis, Free Radic Biol Med 169 (2021) 271-282.
[2] J.Y. Li, H. Tawfeek, B. Bedi, X. Yang, J. Adams, K.Y. Gao, M. Zayzafoon, M.N. Weitzmann, R. Pacifici, Ovariectomy disregulates osteoblast and osteoclast formation through the T-cell receptor CD40 ligand, Proc Natl Acad Sci U S A 108(2) (2011) 768-73.
[3] W. Zhang, W. Hou, M. Chen, E. Chen, D. Xue, C. Ye, W. Li, Z. Pan, Upregulation of Parkin Accelerates Osteoblastic Differentiation of Bone Marrow-Derived Mesenchymal Stem Cells and Bone Regeneration by Enhancing Autophagy and beta-Catenin Signaling, Front Cell Dev Biol 8 (2020) 576104.
[4] C. Ye, W. Hou, M. Chen, J. Lu, E. Chen, L. Tang, K. Hang, Q. Ding, Y. Li, W. Zhang, R. He, IGFBP7 acts as a negative regulator of RANKL-induced osteoclastogenesis and oestrogen deficiency-induced bone loss, Cell Prolif 53(2) (2020) e12752.
[5] W. Yang, X. Lu, T. Zhang, W. Han, J. Li, W. He, Y. Jia, K. Zhao, A. Qin, Y. Qian, TAZ inhibits osteoclastogenesis by attenuating TAK1/NF-kappaB signaling, Bone Res 9(1) (2021) 33.
[6] A.Y.H. Ng, Z. Li, M.M. Jones, S. Yang, C. Li, C. Fu, C. Tu, M.J. Oursler, J. Qu, S. Yang, Regulator of G protein signaling 12 enhances osteoclastogenesis by suppressing Nrf2-dependent antioxidant proteins to promote the generation of reactive oxygen species, Elife 8 (2019).
[7] Z. Zheng, X. Zhang, B. Huang, J. Liu, X. Wei, Z. Shan, H. Wu, Z. Feng, Y. Chen, S. Fan, F. Zhao, J. Chen, Site-1 protease controls osteoclastogenesis by mediating LC3 transcription, Cell Death Differ 28(6) (2021) 2001-2018.
[8] Y. Liu, C. Wang, G. Wang, Y. Sun, Z. Deng, L. Chen, K. Chen, J. Tickner, J. Kenny, D. Song, Q. Zhang, H. Wang, Z. Chen, C. Zhou, W. He, J. Xu, Loureirin B suppresses RANKL-induced osteoclastogenesis and ovariectomized osteoporosis via attenuating NFATc1 and ROS activities, Theranostics 9(16) (2019) 4648-4662.
[9] X. Sun, Z. Xie, B. Hu, B. Zhang, Y. Ma, X. Pan, H. Huang, J. Wang, X. Zhao, Z. Jie, P. Shi, Z. Chen, The Nrf2 activator RTA-408 attenuates osteoclastogenesis by inhibiting STING dependent NF-kappab signaling, Redox Biol 28 (2020) 101309.
[10] D. Laha, M. Deb, H. Das, KLF2 (kruppel-like factor 2 [lung]) regulates osteoclastogenesis by modulating autophagy, Autophagy 15(12) (2019) 2063-2075.
[11] S.Y. Han, J.H. Kim, E.H. Jo, Y.K. Kim, Eleutherococcus sessiliflorus Inhibits Receptor Activator of Nuclear Factor Kappa-B Ligand (RANKL)-Induced Osteoclast Differentiation and Prevents Ovariectomy (OVX)-Induced Bone Loss, Molecules 26(7) (2021).
[12] R. Inai, R. Nakahara, Y. Morimitsu, N. Akagi, Y. Marukawa, T. Matsushita, T. Tanaka, A. Tada, T. Hiraki, Y. Nasu, K. Nishida, T. Ozaki, S. Kanazawa, Bone microarchitectural analysis using ultra-high-resolution CT in tiger vertebra and human tibia, Eur Radiol Exp 4(1) (2020) 4.
[13] Q. Geng, H. Gao, R. Yang, K. Guo, D. Miao, Pyrroloquinoline Quinone Prevents Estrogen Deficiency-Induced Osteoporosis by Inhibiting Oxidative Stress and Osteocyte Senescence, Int J Biol Sci 15(1) (2019) 58-68.

Round 2
Reviewer 2 Report
Thanks for the authors' cover letter.
They answered my all questions.